# Decentralized System Synchronization among Collaborative Robots via 5G Technology

**DOI:** 10.3390/s24165382

**Published:** 2024-08-20

**Authors:** Ali Ekber Celik, Ignacio Rodriguez, Rafael Gonzalez Ayestaran, Sirma Cekirdek Yavuz

**Affiliations:** 1Department of Computer Engineering, Yildiz Technical University FBE, Istanbul 34220, Turkey; smyavuz@yildiz.edu.tr; 2Department of Electrical Engineering, University of Oviedo, 33203 Gijon, Spain; rayestaran@uniovi.es

**Keywords:** distributed system, synchronization, ordering, cloud, 5G, network performance

## Abstract

In this article, we propose a distributed synchronization solution to achieve decentralized coordination in a system of collaborative robots. This is done by leveraging cloud-based computing and 5G technology to exchange causal ordering messages between the robots, eliminating the need for centralized control entities or programmable logic controllers in the system. The proposed solution is described, mathematically formulated, implemented in software, and validated over realistic network conditions. Further, the performance of the decentralized solution via 5G technology is compared to that achieved with traditional coordinated/uncoordinated cabled control systems. The results indicate that the proposed decentralized solution leveraging cloud-based 5G wireless is scalable to systems of up to 10 collaborative robots with comparable efficiency to that from standard cabled systems. The proposed solution has direct application in the control of producer–consumer and automated assembly line robotic applications.

## 1. Introduction

Factory automation is a key pillar of Industry 4.0, which involves the use of advanced control systems, machinery, and information technologies to streamline and optimize manufacturing processes with the primary goal of minimizing human intervention, reducing errors, and increasing the speed and precision of manufacturing operations [1]. In this respect, collaborative robots (cobots) are a significant innovation, representing a shift from traditional industrial robots that typically operate in isolation. Unlike their predecessors, cobots are designed to safely work alongside human operators, fostering a collaborative and flexible manufacturing environment. Industrial work cells implementing cobots (or systems of cobots) are typically controlled from centralized programmable logic controllers (PLCs), which comes with the associated benefit of unified control, fostering seamless communication, coordination, and synchronization, ensuring a harmonized workflow [2]. However, this also imposes a number of challenges and limitations, such as single-point-of-failure, limited scalability, complex integration, costly and time-consuming upgrades, or adaptability to dynamic re-configurable environments, which might lead to significant delays and disruptions in the production process [3]. In this scenario, decentralized control strategies would be better suited, as they allow one to distribute decision-making among the robots, with each robot operating autonomously or in collaboration with its peers, enabling robust, scalable, and re-configurable operation at the cost of slightly increased complexity [4].

Here, the convergence of cloud computing and 5G technology comes to hand, enabling the potential of providing reliable and seamless connectivity between cobots without the need of a cabled centralized PLC [5]. These two technologies enable robust possibilities of maintaining consistent and secure control over distributed systems and applications. 5G facilitates rapid failover between geographically dispersed cloud data centers, ensuring continuous operation even in the event of localized failures. Additionally, 5G supports enhanced redundancy by enabling multiple parallel connections, which can be used to back up critical data and processes in real-time. 5G cloud network architectures also aid in resilience against disruptions, as 5G networks can dynamically reroute data to avoid faulty nodes or congested areas. As a result, the integration of 5G into global cloud-based control frameworks not only enhances the performance and scalability of these systems but also significantly improves their overall reliability and stability [6].

### 1.1. State of the Art

Decentralized coordination of robots is a crucial aspect in multi-robot systems to efficiently achieve system-level objectives. Various studies have explored different approaches to decentralized coordination, emphasizing the importance of communication, control, and adaptability among robots. Refs. [7,8] propose decentralized adaptive control methods for multi-robot collaborative manipulation, focusing on load distribution and coordination without direct communication between robots. These methods showcase the effectiveness of decentralized coordination in achieving collaborative tasks efficiently. Ref. [9] emphasizes the fault tolerance and scalability of decentralized multi-robot systems due to their distributed architecture and modular nature, highlighting the robustness and flexibility offered by decentralized coordination in handling various scenarios and system complexities. Ref. [10] sets the focus on scalability and communication between nodes, illustrating the advantages in terms of scalability, robustness, and privacy. In line with this, in [11], a decentralized queuing algorithm for the multi-robot task allocation problem was proposed, allowing the avoidance of communication bottlenecks and disruptions and ensuring efficient utilization of computational resources.

In general, studies in the literature agree on the fact that inter-process communication between nodes in decentralized approaches is paramount as it has an impact on the efficiency of the synchronization/coordination of the operations [12]. Several studies have addressed the challenges and solutions related to time synchronization in robotic systems from a theoretical, analytical point of view. Ref. [13] discusses the importance of time synchronization in robot networks and proposes a quick two-way time message exchange method for achieving synchronization. Ref. [14] focuses on nonlinear control and synchronization with time delays in multiagent robotic systems, presenting a synchronization control law to address time-delay issues in cooperative network communication. Ref. [15] introduces a general and efficient system for precise sensor synchronization in robotic computing, highlighting the challenges faced in time synchronization within robotic workloads. Furthermore, ref. [16] presents an approach to cooperative robot control and concurrent synchronization of operations. Ref. [17] develops a synchronized adaptive control strategy to coordinate manipulators with time-varying actuator constraints and uncertain dynamics, enhancing cooperative performance among networked robots. Ref. [18] discusses a control strategy for distributed actuators with compensation for communication delays, emphasizing the importance of time synchronization in addressing communication delays among robot elements.

However, most of the designed algorithms or access schemes with a focus on decentralized control that are reported in the literature assume typically ideal or simplified communication channels. This was due to the fact that, until recently, control networks for robotics were exclusively based on physically cabled technologies, typically leveraging field buses or, more recently, Ethernet [19]. These wired technologies offer high capacity, low latency, and low jitter, and therefore the performance of decentralized synchronization schemes is generally bounded in nominal conditions. For example, the access delay reported in [20] for three different decentralized control Ethernet-based case studies was in the range of 3–10 ms, considering a system with up to 50 nodes. Over Ethernet, this represents only a small increase in access time in comparison with centralized strategies, which exhibit a reference access time performance of 0.3–4.2 ms for a single agent under different control protocols [21].

With the advent of Industry 4.0, wireless technologies such as Wi-Fi and 5G are beginning to play an important role in connection to robot control. These technologies are key elements, not only for the support of mobile robotic elements (which cannot implement control cabling for obvious mobility reasons) but also to replace cabling, enabling flexibility and reconfigurability in certain robotic production systems within factories [22]. Here, coordinated access algorithms designed assuming ideal communications might find limited applicability due to a lack of validation and performance evaluation over non-deterministic wireless communication links. There are few studies addressing performance evaluation of decentralized coordination solutions over wireless. Ref. [23] details the performance of three different distributed control architectures for mobile robotics utilizing Wi-Fi, achieving a coordination time performance of 0.2–0.6 s for a single robot. Also with a focus on Wi-Fi, ref. [24] reported 2.2–5.6 s access times in a multi-agent robotic system with three robots. With a focus on 5G, in ref. [25], a multi-agent scheme was proposed for digital twin purposes, with an average task execution time of 8.09 s for three nodes.

In particular for the decentralized coordination and control in operational conditions, wireless technologies can be further leveraged when combined with cloud computing technology. As described by the conceptual model and decentralized cyber-physical system (CPS) operation mechanism described in ref. [26], it is possible to utilize a cloud-based agent approach to create an intelligent collaborative environment for product creation. This idea is also supported by the high-level concept and simulations in ref. [27], which demonstrated that cloud platforms serve as a powerful centralized infrastructure for implementing global coordination logic in robotic systems by deploying orchestration software (SW) in combination with reliable 5G communication.

### 1.2. Novelty and Contributions

In this work, we focus on the novel utilization of 5G and cloud technologies to provide decentralized real-time coordination between robotic agents. We leverage 5G for overall system connectivity and cloud infrastructure as a mere message relay entity, keeping all the logic and intelligence distributed across the robotic nodes. Our solution proposes the use of an access method with both causal message ordering and non-causal ordering possibilities that enables decentralized synchronization among the robotic entities, with applicability in different industrial manufacturing applications. Apart from solving the limitations of a centralized control solution and reducing bottlenecks in the development of work cells in industrial automation scenarios, the proposed solution comes with other operational benefits, such as simplified configuration and direct SW execution of the robots, which is of high relevance to cobot consumers nowadays [28].

The reference applicability scenarios that will benefit from the proposed distributed decentralized synchronization solution are the following:Producer–consumer applications [29]: These involve a system where “producers” generate or supply data, materials, or products, and “consumers” utilize or process these resources. This concept is central to many industrial processes, particularly in manufacturing, where automated material handling is of relevance. A cobot-based implementation of this is illustrated in Figure 1. Here, the cobots on the left and right take the roles of “producer” and “consumer”, respectively. Both cobots share a physical area of the production (critical section), where synchronized actions are needed in order to avoid collision accidents or malfunctions of the underlying manufacturing process. In a coordinated and ordered manner, the producer cobot picks the material (cylinder) up from its predefined position on the left and places it within the critical production area. Sequentially, the consumer cobot will access the critical area and pick the cylinder up to place it at its goal position on the right. In this case, non-simultaneous access and operation over the critical shared area and synchronization of actions between consumer and producer, as well as the causal order of operations, are important to ensure that the materials are operated in the correct order set by the production control.Automated assembly line applications [30]: These often involve multiple robots of the same type working together to enhance efficiency and productivity. In scenarios where different robots pick up pieces of material, coordination and precision are key. These scenarios are present in processes in many different industries: car body assembly in car manufacturing, PCB assembly in electronics manufacturing, or bottling lines in the food and packaging industry. In the cobot-based implementation exemplified in Figure 2, the two cobots can pick materials (cylinders) from the critical section area and place them outside. As in the producer–consumer case, the access to the shared production area needs to be coordinated to avoid collisions, but in this case the order of operations is not relevant as any cobot can pick any available cylinder. Thus, the only production control requisite is to ensure non-simultaneous access and updated information about the remaining materials within the critical section.

Specifically, the contributions in this work can be summarized as follows:Design, mathematical formulation, and SW implementation of a multicast-based decentralized synchronization method.Integration, operation, and validation of the decentralized coordination solution over 5G and edge-cloud technology.Scalability testing and performance evaluation of the proposed solution.Comparison of the performance of the proposed solution with that achieved when the solution is implemented over traditional reference control architectures with cabled technology.

The rest of the paper is organized as follows. Section 2 describes the proposed method for decentralized coordination. Section 3 details the implementation considered for validation and evaluation of the proposed solution, including multiple network configurations, allowing the benchmark of the performance achieved over 5G wireless and cloud with respect to the one achieved over other traditional cabled reference control systems. Results are described and discussed in Section 4, and Section 5 addresses operational validation and future research prospects. Finally, conclusions are drawn in Section 6.

## 2. Proposed Multicast-Based Decentralized Synchronization Method

To enable decentralized synchronization among a group of processes (in this case, one process is equivalent to one cobot), a simple causal multicast method based on vector clocks is proposed. Vector clocks are important to ensure causality while maintaining compacted message sizes and scalability as compared to standard Lamport logical clocks [31]. By keeping a “timestamp” vector for each process, where each timestamp is based on sequence numbers, vector clocks are capable of providing information about the causal relationships between events in a distributed system and help track the relative order of events between processes. Another positive aspect of using vector clocks is that, during implementation, they are managed by the processes themselves and are, therefore, transparent to the underlying applications [32].

The proposed decentralized synchronization scheme is defined in general terms in Algorithm 1. Initially, a group *I* of $i=1,\;\ldots ,\;n_{r}$ robot interfaces is defined, where $n_{r}$ represents the number of processes/cobots considered within the distributed system. Each of the interfaces subscribes to a common multicast notification topic to guarantee message (msg) exchange between all decentralized components, and will also initialize its local vector clock (*L*), process queue (*Q*), and message buffer (*B*). Please note that *L* is two-dimensional, as each interface *i* will keep its own vector clock with the same number of elements, $j=1,\;\ldots ,\;n_{r}$. The global critical section (CS) is also initialized. In this boot-up phase, for each interface *i*, *L* is initialized to 0, while all other elements, *Q*, *B*, as well as the global CS, are initialized as empty sets (∅).

After initialization, each of the decentralized robot entities (*p*) will be operating in parallel in a main loop, where they check for incoming multicast messages (multicast.rcv). These messages (msg) carry information about the CS status, (received) vector clock (*R*), id of the sender interface (pid), and type of operation. The size of the vector clocks is determined by $n_{r}$. When a process sends a message (multicast.snd), the value of its index in the vector clock ($R_{p}$) is incremented by one and appended to the sent message. When a message is received, first, causality is checked (line 8), where the received vector clock, *R*, is compared to the local clock, *L*. If causality conditions are met, then the local vector clock, *L*, is updated, and the message processed. If causality conditions are not met, the message is stored in the process queue, $B_{p}$, for later verification (line 36).
**Algorithm 1:** Proposed decentralized synchronization algorithm based on multicast message exchange.
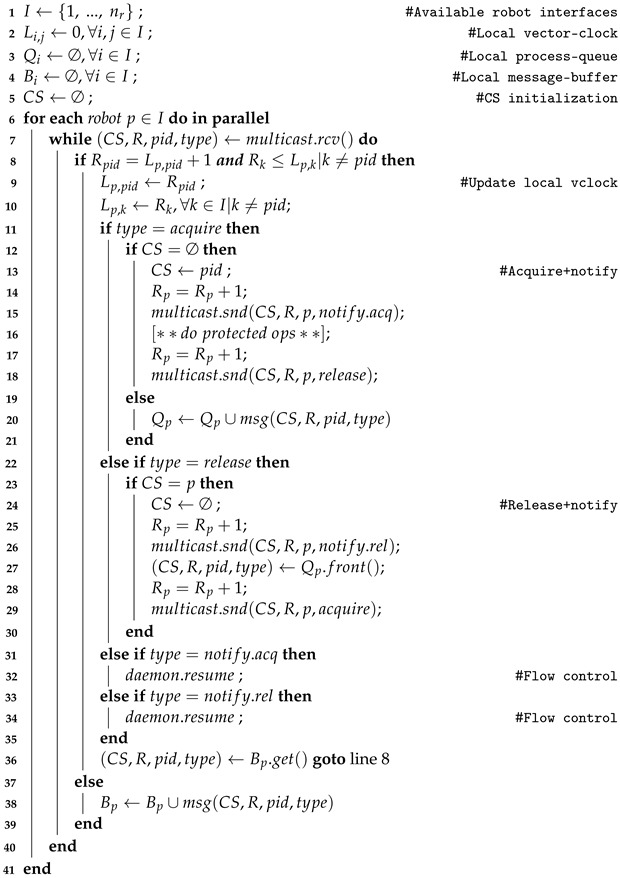



When an *acquire* message is received (line 11), it is firstly checked whether the critical section, CS, is in use. If the CS is in use, the acquire request is queued in $Q_{p}$; otherwise, the lock is acquired by the process, and a notification is issued to the multicast group with a *notify.acq* message. Once the robot has been finalized to perform its protected operations, the CS release procedure is triggered by issuing a release message to the multicast group. When a *release* message is received (line 22), the owner of the CS lock releases the lock and notifies the multicast group with a *notify.rel* message. It also checks in the $Q_{p}$ queue which interface is the following CS lock owner and signals it to the multicast group with an acquire message. When *notify.acq* or *notify.rel* notification messages are received (lines 31 and 33, respectively), an internal background daemon for flow-control is triggered to perform optimization actions, avoiding extra competition (and message exchanges) between interfaces when the CS is in use. Once any message is processed, a new message is retrieved from the $B_{p}$ buffer (line 36) and execution continues.

It should be highlighted that a secondary background daemon is implemented from any active CS–owner interface that issues periodic keep-alive messages to the multicast group through a parallel notification topic. This allows monitoring the connectivity status and identify potential robotic malfunctions, serving as a reliable mechanism to fulfill operational safety regulations and potentially establishing automated recovery mechanisms. The time and space complexities of this algorithm were theoretically evaluated, resulting in $O({n_{r}}^{2})$ and $O(n_{r})$ scenarios, respectively. The time complexity is $O({n_{r}}^{2})$ due to the bi-dimensionality of the vector clock ($L_{i,j}$), while the space complexity is $O(n_{r})$ because the algorithm makes use of local array-based vector clocks, process-queues, message-buffers, and critical section state flags. Overall, the proposed algorithm has a polynomial time complexity and linear space complexity in terms of the number of robot interfaces.

## 3. Network Implementation

In order to evaluate the performance of the proposed decentralized synchronization algorithm, the network architecture displayed in Figure 3 was implemented. It is composed of three main nodes (a local machine, a factory machine, and one instance of the global MS Azure Cloud), with the computing capabilities summarized in Table 1, and two different access networks (5G and Ethernet-LAN), with the reference connection performance numbers compiled in Table 2 for average upstream (US) and downstream (DS) data-rates and round-trip time (RTT) latencies. Message queuing telemetry transport (MQTT) with publish/subscribe (Pub/Sub) architecture was chosen as the high-level communication protocol based on the findings from our previous work [33]. In this architecture, the local machine is in charge of executing a variable number of robot instances, which are publishers/subscribers to a common specific MQTT topic. The MQTT “broker”, acting as message relay, is deployed either at the MS Azure global Cloud instance or at the factory machine, depending on the exact test scenario considered. The following configurations with varying network setups and access conditions were evaluated:5G: This considers that each of the robots is equipped with a 5G modem and an active subscription that allows, in this case, connectivity towards the global MS Azure cloud over a public 5G network. This is the main target of our evaluation as it fully fulfills all decentralization and flexibility requirements from future cobot scenarios. In our practical implementation Quectel RM502Q-AE, modems [34] and 5G IoT enterprise subscriptions with dedicated access point name (APN) from the telecom operator Telenor DK were used. These are depicted in Figure 4. Reference access network conditions are summarized in Table 2.Ethernet (ETH): This configuration requires that the robots are Ethernet-cabled to the factory network for Internet and cloud access, thus limiting the flexibility of the implementation. It is mainly used as a reference to compare the performance of the algorithm over 5G against the one achieved over standard cable Internet connections. As detailed in Table 2, in this scenario, while data-rates are comparable, the RTT latency towards the MS Azure global cloud is lower than over 5G.Local: This represents a configuration with no external cloud dependence. In this case, the broker is deployed locally in the factory machine and robots are Ethernet-cabled to it. This would be equivalent to running the proposed decentralized coordination algorithm on a local line controller.PLC: This case resembles the current situation in typical factory robot work lines, where single-robot control operations are fully centralized and scheduled by a PLC, and further PLC–PLC coordination is needed for multi-robot operations. This configuration is representative of non-decentralized access and is affected by all the operational limitations described in Section 1.

As introduced in Section 1, for the proposed scheme, the size of the messages increases proportionally with the number of agents (robots) to synchronize. The configured MQTT payloads are composed of a mixture of “int” and “string” components, where the most relevant element in this evaluation, the vector clock itself, is int-based. This induces maximum MQTT packet sizes of 203, 238, 263, 313, 334, and 374 B, for configurations with 1, 5, 10, 20, 30, and 40 robots, respectively. With these application layer packet sizes, and considering the 64 B for the headers of the underlying transport, network, and medium access control (MAC) layers, layer 2 frame sizes vary from 267 to 438 B, which translates into approximately 1/6 to 1/3 of the 1500 B Ethernet maximum transfer unit (MTU). No additional vector compaction techniques [35] are applied since the intended application targets small groups of processes, and therefore smaller gains will be achieved at the expense of increased processing complexity. Accounting for another 20 B for physical layer headers, and considering the maximum number of robots for the different configurations, the maximum instantaneous DS/US data-rates will oscillate between 2.30 kbps for a single robot to 146.56 kbps for the more demanding configuration with 40 simultaneous robot instances. It should be noted that, in this implementation, all robot entities share common 5G and Ethernet network interfaces, while in a real deployment each robot instance will have its own individual access interface. However, as per the access network assessment performed prior to the performance evaluation, as summarized in Table 2, and the calculated maximum expected data-rates for the synchronization application, the different access networks (5G and Ethernet) will dedicate well below 1% of their capacity resources. This ensures that the results of the performance assessment will be representative of a non-limited capacity situation with 5G/Ethernet access from multiple simultaneous interfaces, where the latency performance is the main contributor to the coordination method observed in the application performance.

For further clarity on the overall implemented network system model, Table 3 summarizes the main components and characteristics of all the different examined combinations.

### Key Performance Indicators and Data Processing

Performance measurements were executed for the described configurations with variable numbers of robots (1, 5, 10, 20, 30, and 40). The scalability testing evaluation focused on benchmarking the time taken by the “acquire” ($t_{a}$) and “release” ($t_{r}$) actions to succeed in operating the distributed mutexes that control access to the critical section, as they are representative of the efficiency of the proposed scheme in providing coordination and communicating overall system logic status to all decentralized nodes. Both timings are measured and logged locally at each local robot controller following the execution of Algorithm 1. In particular, $t_{a}$ is evaluated at line 13 ($t_{line13}$), while $t_{r}$ is assessed at line 24 ($t_{line24}$). Both variables are examined over local node runtime, with starting time set at line 6 ($t_{0}=t_{line6}=0\;$s). Therefore, $t_{a}$ and $t_{r}$ are computed as defined in Equations (Equation 1) and (Equation 2), respectively.
(1)$$t_{a}\;\left[s\right]=\left\{\begin{array}{cc} t_{line13}-t_{0} & \mathrm{if}\;\mathrm{first}\;\mathrm{CS}\;\mathrm{acquisition}\\ t_{line13}-t_{line24} & \mathrm{otherwise} \end{array}\right.$$
(2)$$t_{r}\;\left[s\right]=t_{line24}-t_{line13}$$

With the aim of focusing the analysis only on the access performance of the proposed scheme and the impact of the different communication technologies, the “protected ops” implemented at line 16 is a simple counter that monitors the number of CS accesses ($\#CS_{a}$) at a given node. Thus, the operation executed for evaluation is the one describe by Equation (Equation 3), being $\#CS_{a}=0$ at the beginning of the runtime execution at each of the interfaces. This operation is not very demanding, and its effective runtime execution time is in the order of 0.5–1 ms.
(3)$$\#CS_{a}=\#CS_{a}+1$$

The overall decentralized access performance can be quantified as total access time ($t_{t}$), computed as the sum of “acquire” and “release” times, as indicated in Equation (Equation 4).
(4)$$t_{t}\;\left[s\right]=t_{a}+t_{r}$$

It should be noted that, when the proposed scheme is put into production, the control instructions for a given cobot will be block coded and executed within the “protected ops” and, therefore, the overall $t_{t}$ will be increased accordingly to the duration of the implemented operations within the physical industrial production operation domain. However, as the focus of this study is set purely on communication and synchronization aspects, the simple counter increase operation with negligible runtime duration is considered.

For each of the case experiments, a total performance test duration ($t_{d}$) of 300 s (5 min) was considered, where the system was configured to perform continuous CS access attempts according to the proposed scheme and variable target number of robots. For further insight on the performance, measurements are executed for the following:Proposed causal ordering method (as per Algorithm 1). This would be representative of industrial applications such as the producer–consumer use case described in Section 1.2.Non-causal random access, where robots individually access the critical section but no order/coordination is guaranteed within the process (same baseline execution, except that the causal access conditions at line 8 in Algorithm 1 are skipped, triggering that CS “acquire” requests are processed as they come and allowing that the CS is acquired by the quickest interface without any specific synchronized order). This would be representative of applications similar to the automated assembly line scenario described in Section 1.2.

This separate evaluation allows one to further evaluate the overhead cost in the execution time of executing an ordered sequence over randomized access (Δ) by computing the difference in total average execution time, as defined in Equation (Equation 5).
(5)$$\Delta \;\left[s\right]=t_{t,non\text{-}causal}-t_{t,causal}$$

The performance results are also briefly examined from an industrial system perspective. Two metrics are defined with the objective of estimating the efficiency of the access schemes over the different network technology solutions in terms of robotic cell speed (λ) and production cycle time (PCT). λ is evaluated by computing the rate of CS accesses during the different tests, as defined in Equation (Equation 6), providing a useful reference for the estimation of the achievable number of robotic operations within given industrial operation periods. Equation (Equation 7) is used to estimate the overall operational time taken by the system coordination functions, based on the number of robots composing the cell and the median total access times for given schemes and network configurations.
(6)λ#CSamin=#CSatd
(7)$$PCT\;\left[s\right]\approx n_{r}\cdot t_{t}$$

While λ provides a reference of the instantaneous capacity of the robotic cell, PCT represents an estimation of the total cycle execution time assuming a robotic cell where all robots are configured to execute one access. Therefore, both parameters are inversely proportional: for large capacity systems, the resulting cycle time will be small. This has a further implication on industrial production throughput, as fast cycle times are directly related to high production levels.

## 4. Performance Results and Discussion

An extensive analysis and comparison of the performance evaluation results obtained for the different configurations is detailed in the following section.

### 4.1. Acquire, Release, and Total Access Times

Figure 5 depicts the measurement results for the time duration of the “acquire” and “release” actions for the proposed coordinated causal ordering scheme considering the different network access configurations. To ease understanding, each set of results is color-coded, following the same color schemes used in Figure 3 for each of the access topologies: 5G in green, ETH in black, local in magenta. The different sub-figures summarize the statistics of $t_{a}$ and $t_{r}$ in the shape of boxplots, where the boxes indicate performance results bounded within the 25–75%-iles, and the middle line indicates the median value of the distributions (50%-ile). The lower and upper whiskers indicate values at the 1%-ile and 99%-ile, respectively.

As expected, $t_{a}$ increases with an increasing number of robots for both the causal ordering and the non-causal case for all 5G, ETH, and local configurations. The cabled local configuration with the MQTT broker deployed in the factory machine presents the best performance, bounded below 10 s, except when 40 robots are connected. In this case, the ETH access configuration with wired Internet connection and MQTT broker hosted by the MS Azure global cloud exhibits a lower $t_{a}$, with a median of 12 s. The 5G configuration presents the highest acquisition times for all number of robots. This was also expected, as per the latency performance values summarized in Table 2. The capabilities of 5G wireless are, from the design, due to the air media transmission characteristics, more limited in terms of latency and capacity than the ones from cabled technologies, but they are more beneficial in terms of flexibility and re-configuration of the machines in production [36]. For the 5G configuration, $t_{a}$ is bounded by 5 s for up to 10 robots, increasing up to 10–36 s for 20–40 robots. As the “release” action is performed immediately after one robot finalizes its interaction with the CS, its performance is almost instantaneous for all configurations and numbers of robots. In general, $t_{r}$ is lower than 0.25 s, except for the local configuration with 40 robots and the 5G configuration with more than 20 robots. Overall, 5G exhibits a slightly larger $t_{r}$ than the other technologies, but it is still bounded, which is motivated by the highest variability in the access media, as explained for $t_{a}$.

The performance results for “acquire” and “release” actions for the uncoordinated non-causal case are displayed in Figure 6, organized in a similar fashion as the previous results presented for causal execution. Similar trends are observed for this non-causal case as compared to the causal case for both “acquire” and “release” times, but with lower absolute performance values. The cabled ETH and local configurations exhibit better performance as compared to the 5G one at the expense of the aforementioned operational limitations. For up to 30 robots, $t_{a}$ is bounded by 9 s for ETH and local, increasing slightly up to 14 and 19 s, respectively, for the 40-robot case. For the 5G configuration, $t_{a}$ is well bounded below 5 s up to 10 robots, increasing to up to 10, 17, and 30 s for 20, 30, and 40 robots, respectively. In terms of $t_{r}$, performance is shown to be well below 0.25 s for most of the configurations, except those with 40 robots in the ETH and local cases and those with more than 20 robots in the 5G one. On average, the uncoordinated non-causal access performance for $t_{r}$ does not present any significant increase as compared to the coordinated causal case. For the acquisition time, the situation is different. For the 5G case, while up 20 robots, $t_{a}$ is less than 0.5 s faster in the non-causal case than in the causal case, for 30–40 robots, uncoordinated access is 1.1–4.2 s faster than the coordinated one. In the ETH case, both $t_{a}$ and $t_{r}$ exhibit a comparable performance for both the non-causal and causal access schemes. In the local case, coordinated and non-coordinated access performs similarly, except for the 40-robot configuration, where a 2.1 s improved performance is observed for $t_{a}$ for the uncoordinated non-causal schemes.

The performance in terms of median total access time, including the overall effect of the “acquire” and “release” accesses for the different configurations and schemes, is examined in Figure 7. As a reference for discussion, a vertical dashed line is included at 5 s, illustrating the maximum access time expected in typical PLC-based industrial systems [37]. This is the time margin allowed in delay-tolerant systems for acknowledging the reception of control messages and keep-alive communications in centralized settings [36], which can be used as a threshold for our decentralized approach to determine which of the evaluated cases will perform in a comparable fashion to traditional industrial systems. In our case, if the total access time is confined within said threshold, it would mean that timely active communication is being established between all robots in the system, and thus the industrial system is performing nominally.

As described in the figure, and briefly addressed previously, the performance trends for the causal and non-causal access schemes are very similar, with minor differences for configurations of up to 20 robots, and bounded differences of up to 4.2 s in the 30- and 40-robot cases. These are further elaborated in Section 4.2. As $t_{r}$ was in all cases very low (typically below 0.5 s), $t_{t}$ is dominated by $t_{a}$, and thus increasing with the number of robots for all decentralized configurations (5G, ETH, and local). The figure also includes the performance measured for the traditional centralized PLC case, which exhibits a very low and constant performance, with total access time values of approximately 0.2 s, which is conditioned by all limitations elucidated in Section 1. In general, the total access time performance for the 5G, ETH, and local configurations is similar to the centralized PLC one for the single robot case. As the number of robots to be coordinated in the industrial system increases, the difference from the decentralized schemes increases is more apparent. For five robots, $t_{t}$ is 4–9 times larger for the decentralized configurations as compared with the PLC case, increasing to 9–20, 23–46, 42–70, and 89–170 times for 10, 20, 30, and 40 robots, respectively. When putting the results in the perspective of the considered reference PLC survival time of 5 s, it is observed how all tested decentralized 5G, ETH, and local configurations present total access times below the reference threshold for robotic systems with up to 10 robots, exhibiting comparable capabilities to centralized PLC-controlled systems. Above that number of robots, the communication control timer would be exceeded, and thus the proposed synchronization scheme would underperform in comparison with traditional PLC-based cabled systems.

In particular for the decentralized causal synchronized method over the 5G and cloud network settings, the full statistics of its combined “acquire”–“release” performance are shown in Figure 8, for both the coordinated causal and uncoordinated non-causal access schemes, in terms of the empirical cumulative distribution functions (ECDF) of the overall access time. It is observed that for up to 20 robots, $t_{t}$ is bounded below 10 s for up to 20 coordinated robots, with similar performance for the causal and non-causal 5G cloud access schemes. It is also noticed that, for up to 30 robots, the overall access is quite deterministic, with a low dispersion or deviation around the median values. Moreover, in those settings, the median total access time follows a linearly increasing time, with the total number of robots with tt≈12·#robots. Above that, for 40 robots, $t_{t}$ presents larger variations, spanning over multiple tens of seconds, and an increased linear growth rate with median values of tt≈34·#robots. For the 30- and 40-robot configurations, the access time performance over the 5G cloud is slightly increased when causal ordering is applied, resulting in a 1.1–4.2 s slower access as compared to the non-causal case, as previously discussed in the above. As indicated, the reported total access time performance over 5G would fulfill the reference survival time requirements for configurations with up to 10 synchronized robots. For larger number of robots, $t_{t}$ begins to exhibit a quadratic increase behavior, imposed by the multicast-based notification method considered in the algorithmic implementation, and greatly impacted by the global cloud access RTT over 5G, which is approximately four times larger than over cabled Ethernet, as summarized in Table 2. For up to 10 robots, despite the approximately 10 times higher access time as compared to using traditional cabled PLC-based centralized control schemes, our wireless 5G cloud solution exhibits bounded reliable performance. This will enable flexibility and re-configurability within the industrial setup, which will further lead to potential operational production gains.

To complete the analysis of the 5G cloud configuration, key total access time performance values are summarized in Table 4, together with those observed in the comparable cabled ETH and local scenarios. The shaded cells highlight those configurations for which the performance satisfies the 5 s maximum communication timers configured in traditional industrial control systems. These results further emphasize the comparable performance of the decentralized coordination method over 5G wireless with that from other cabled network solutions for configurations with up to 10 robots. They also illustrate the limitations of the current decentralized coordination solution in terms of scalability for robotic systems with more than 10 robots. However, this is not considered as a problem, as the current solution would suffice the needs in small to medium-sized manufacturing systems, typically devoted to specific tasks like welding, painting, or assembly, often used in smaller production lines where specific tasks can be automated to improve efficiency and consistency, as these are the key target settings for wireless-based automation to induce re-configurability and flexibility within the industrial manufacturing process [38].

In comparison to the reported literature in Section 1.1 addressing performance evaluation of decentralized coordination solutions, for a single robot, our 5G cloud solution presents similar performance to the Wi-Fi one detailed in [23]. For multi-robot configurations, our 5G cloud solution outperforms the Wi-Fi one reported in [24]. While their solution exhibited access times of 2.2–5.6 s with 3 robots, ours is capable of providing such access time levels for configurations with up to 10 robots. Along similar lines, as compared to the 5G decentralized 5G solution described in [25], with an average execution time of 8.09 s for three nodes, our proposed method shows approximately eight times improved performance, being able to coordinate up to 20 robots with similar performance reference levels.

### 4.2. Causal vs. Non-Causal Decentralized Access

The total access time performance results can be further leveraged to evaluate the impact of the extra processing included in the proposed decentralized coordination method to ensure causality in the access. Figure 9 examines Δ, which quantifies the overhead in access time experienced when robots operate in a specific order the shared process (causal coordinated solution) as compared to the case where any robot could access the shared process in a random non-pre-defined order (non-causal uncoordinated method). As described in Section 4.1, the difference observed in performance for the decentralized coordinated causal access schemes and the uncoordinated non-causal one was small. Considering median $t_{t}$ performance, the execution time overhead is negligible for the cabled scenarios (ETH and local) and bounded by 0.1 s for 5G, for up to 10 robots. For a larger number of robots, the penalty of providing causal ordering on top of the random access begins to become costly, especially in the 5G and local scenarios, which can reach an access overhead of up to 2.5–4.2 s with 40 robots in the system. In the 5G global cloud case, this is motivated by the 5G network performance, while in the local cabled case it is mainly due to the limited processing power of the considered factory machine for broker operation.

### 4.3. Overall Industrial System Performance

It is also possible to analyze the performance results of the decentralized coordination method from the perspective of the industrial system. In Figure 10, the median number of accesses to the critical section per minute are compared for the different schemes and network configurations. Each access to the CS can be seen as a turn of operation for that particular robot that has obtained the access. Therefore, λ provides an indication of the speed efficiency of a given robotic production cell composed of a variable number of robots coordinated by the different proposed access schemes and network topologies. As λ is tightly dependent on $t_{t}$, the best robot access rate (285 accesses/min) is achievable for the centralized PLC configuration. The only decentralized solution that accomplishes similar access rates for single robot settings is the local one. In this case, ETH and 5G configurations present reduced rates of 257 and 154 accesses/min, respectively. For multi-robot configurations, λ is significantly reduced for the decentralized solutions. For 10 robots, which sets the tolerance performance limit for our proposed solution, the system speeds are reduced to 29 accesses/min for the local configuration, 23 accesses/min for the ETH one, and 14 accesses/min for 5G.

From an industrial manufacturing perspective, the observed performance of the multiple configurations can be translated into production cycle times. In this respect, Figure 11 illustrates the estimated PCT for a given robotic production cell composed of a variable number of robots. As observed, for a robotic cell with a single robot, despite λ being very different for all network access configurations, they all present a very similar PCT (0.2–0.4 s). This makes sense, as with a single robot, all the decentralized 5G, ETH, and local schemes, and also the centralized PLC one, translate into point-to-point control systems where the main access limiting factor and contributor to the overall operation cycle is the network access performance. As described in Section 4.1, robotic cells for up to 10 robots would allow control over 5G in a causal decentralized manner, with comparable efficiency to that from traditional cabled control systems. In this case, a median PCT of 42.7 s is estimated over 5G, with a median robot access rate to the system of 14 robots/min. This implies an overall degradation of 17.1–22.0 s (66–106%) with respect to the decentralized ETH and local configurations, and 40.6 s (1933%) with respect to the traditional centralized PLC configuration. For further reference, performance values for all configurations are summarized in Table 5.

## 5. Validation in Operational Conditions and Future Research Prospects

It should be noted that the reported performance of the decentralized robot synchronization scheme over 5G edge-cloud could be perceived as a big penalty in terms of production performance. However, the estimated 5G PCT performance values can be easily limited or compensated in operational manufacturing systems, as real production plans include other operational service aspects such as planned maintenance stops or line re-configurations, whose duration, typically in the order of several orders of magnitude larger than PCT, will be notably reduced if cables are avoided in the physical industrial system and replaced with wireless 5G [39,40]. It is therefore expected that the benefits of operating robotic coordination over 5G and edge-cloud-based systems, as described in Section 1 and Section 3, will even result in positive production gains when considering the complete range of operational manufacturing-related features.

All results presented in this work have been computed based on the reference network architecture illustrated in Figure 3, which emulates the host node, network, and computing capabilities of real industrial systems. However, the proposed solution has already been tested in real industrial operational conditions. The real setups differed slightly from the ones with two cobots presented in Figure 2. Those were used merely for use-case exemplification, while our proposed solution has been validated in larger real-world systems of up to eight cobots. Due to confidentiality, we cannot show a picture of that specific industrial setup environment. Instead, we share Figure 12, where we display another relevant realistic multi-robot scenario that made use of our developed solution. In this case, our decentralized 5G-based solution was used to coordinate the operations of a stationary robotic cell and two cobots installed on top of autonomous mobile robots. This industrial application scenario is currently being explored in collaboration with Aalborg University, Denmark, and a number of industrial partners, as it represents a reference implementation that demonstrate the key 5G potential for industrial use [36].

This work signifies a first approach towards the integrated use of 5G technology and cloud computing for robust and reliable decentralized control, which has been successfully validated in multi-robot operational conditions based on industrial-grade equipment. However, as illustrated by the performance results in Section 4, the current implementation presents a limitation when the number of interfaces required to coordinate exceeds 10. Future work will consider optimizations in the coordination algorithm, potentially developing enhanced multicast, multi-node communication methods with improved time complexity. Data security has not been explicitly addressed in the current solution, which makes use of MQTT over 5G as an end-to-end application layer communication protocol. Currently, the control messages are not further encrypted, but this will be explored in future implementations. While, by the 5G native design, radio transmissions are fully-encrypted, end-to-end encryption needs to be addressed at a high system level, considering further aspects such as Information Technology/Operational Technology (IT/OT) integration with the physical industrial HW [41].

## 6. Conclusions

Decentralized synchronization among collaborative robots (cobots) can be achieved by leveraging 5G and cloud computing technologies, simplifying the configuration and enhancing the flexibility of current industrial setups. A novel solution was proposed, implemented and validated considering public 5G network access and MS Azure global cloud support for simple message relaying. The performance of the proposed 5G coordination solution was evaluated for scalability considering groups of different numbers of robots and benchmarked against alternative cabled solutions.

The results demonstrate the feasibility of the proposed scheme in providing synchronization and causal ordering over 5G to groups of up to 10 cobots, exhibiting bounded median and maximum access times of 0.4–4.1 and 0.6–5.3 s, respectively. These performance values are comparable to those from current traditional cabled PLC-based industrial systems. The observed performance over public 5G is approximately 40% slower as compared to the same solution over cabled Ethernet Internet access. However, this should not be perceived as negative since the 5G wireless setup presents other associated benefits such as its operational flexibility and re-configurability, which can translate into large production gains in the long-term of the industrial operations. From an industrial production perspective, the results indicate that the achievable production cycles considering cobot cells operating with the proposed 5G cloud solution are similar for applications requiring causal-ordered coordination, such as material handling, or for automated assembly line use cases, which do not, typically, require causality, and thus result in slightly optimized cycle times.

## Figures and Tables

**Figure 1 sensors-24-05382-f001:**
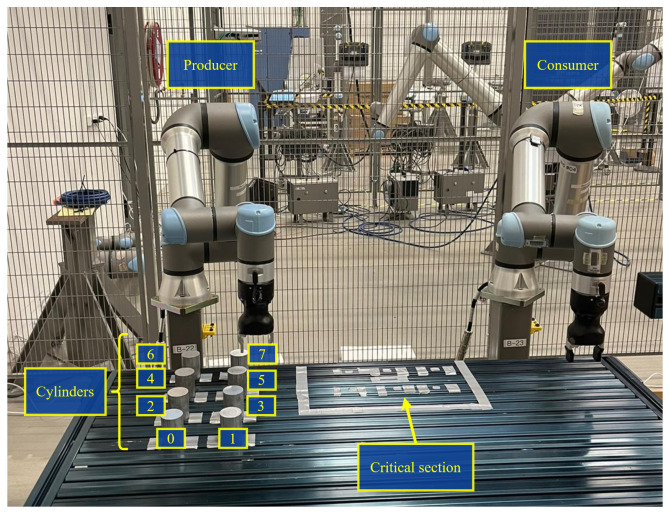
Illustration of a producer–consumer application requiring decentralized cobot coordination with causal ordering.

**Figure 2 sensors-24-05382-f002:**
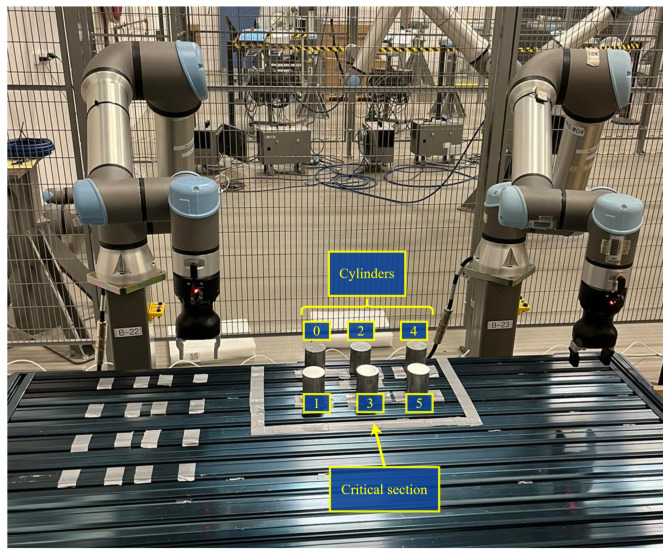
Illustration of an automated assembly line application requiring decentralized cobot coordination and non-causal ordering.

**Figure 3 sensors-24-05382-f003:**
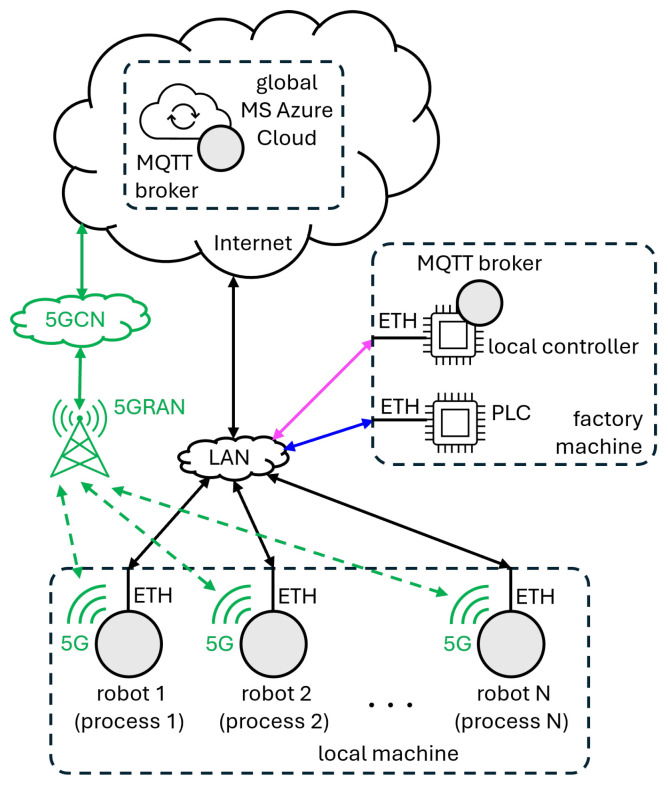
Implemented network architecture, including main nodes (local machine, factory machine, and global cloud) and access networks (5G and Ethernet-LAN). Solid double-sided arrows represent wired connections, while dashed ones specify wireless connectivity.

**Figure 4 sensors-24-05382-f004:**
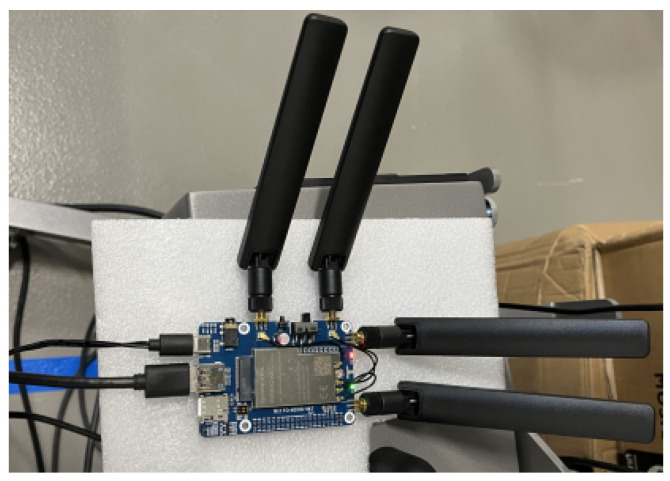
Picture of the modem installed on the local cobot controller (local machine) in the 5G setup for 5G wireless access evaluation.

**Figure 5 sensors-24-05382-f005:**
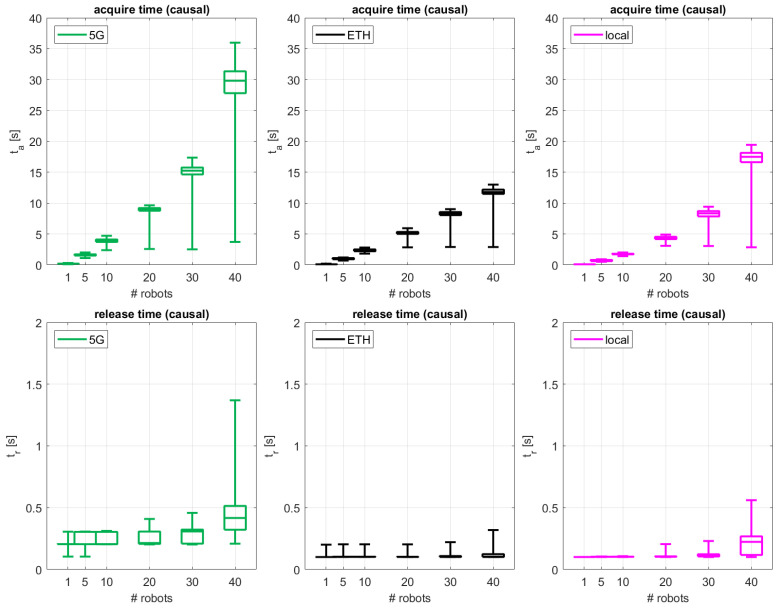
Performance results in terms of acquisition time and release time for the coordinated causal access scheme for the different network configurations.

**Figure 6 sensors-24-05382-f006:**
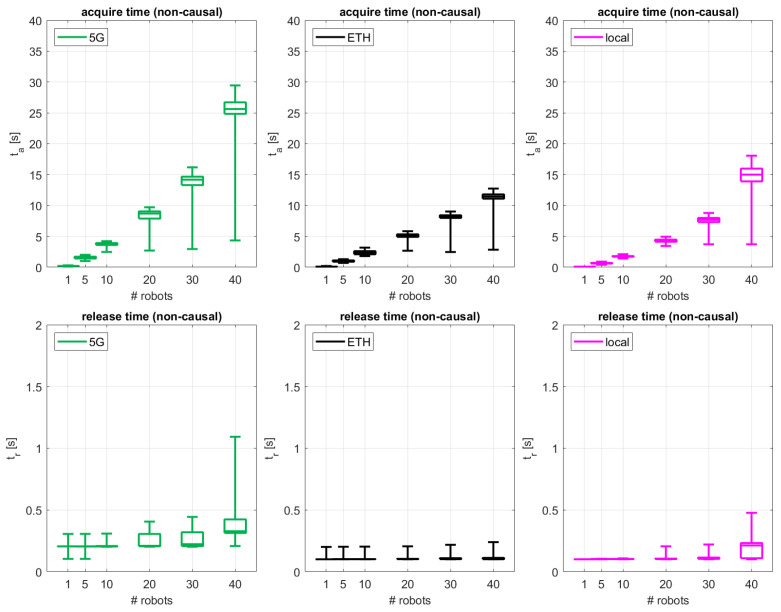
Performance results in terms of acquisition time and release time for the uncoordinated non-causal access scheme for the different network configurations.

**Figure 7 sensors-24-05382-f007:**
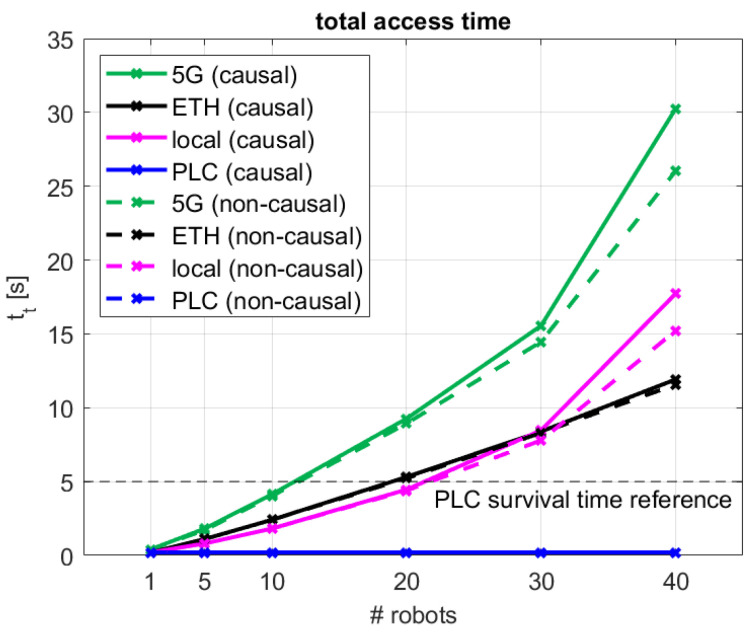
Median total access time for the different network configurations for both the coordinated causal and uncoordinated non-causal access schemes.

**Figure 8 sensors-24-05382-f008:**
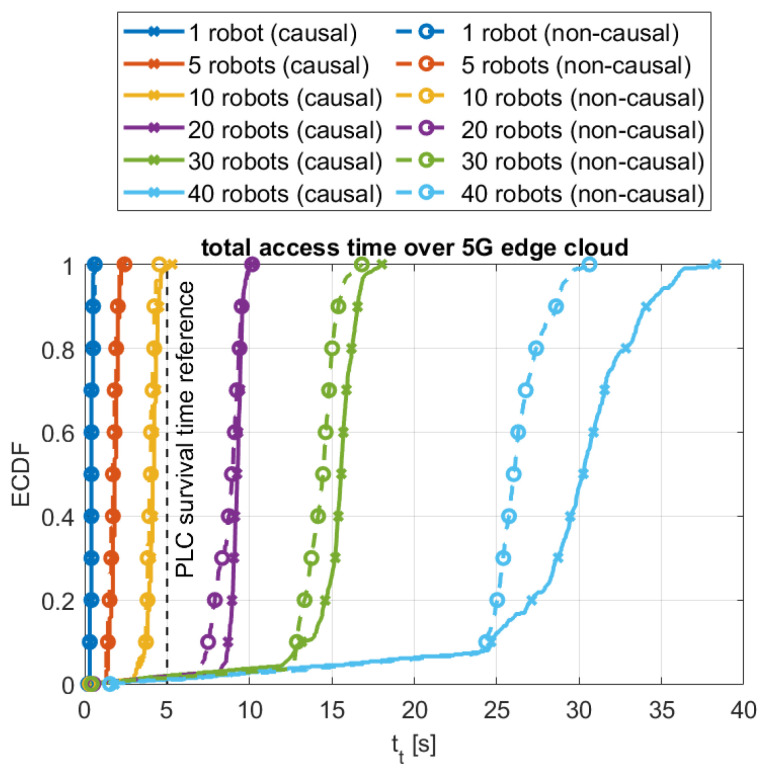
Statisticalsummary of the total access time performance of the decentralized coordination scheme for both the causal access and uncoordinated non-causal configurations considering different numbers of robots.

**Figure 9 sensors-24-05382-f009:**
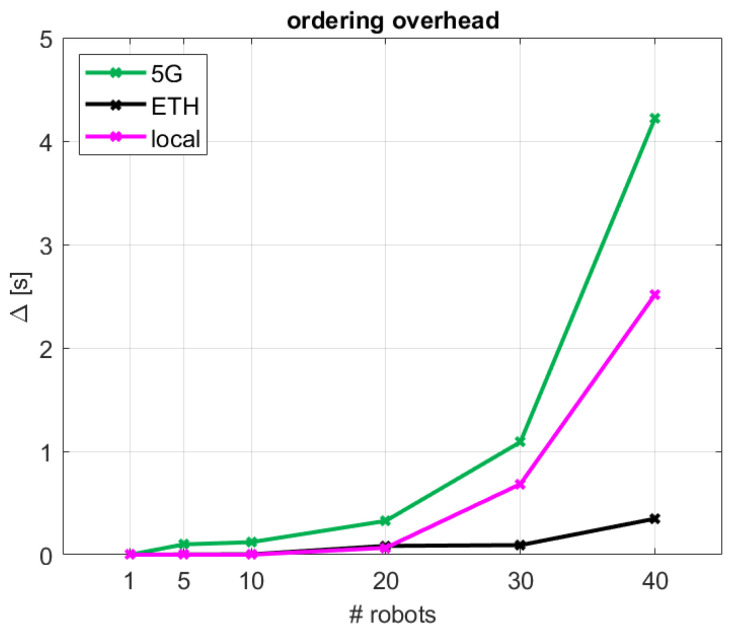
Performance impact of the causal ordering overhead for the different network configurations.

**Figure 10 sensors-24-05382-f010:**
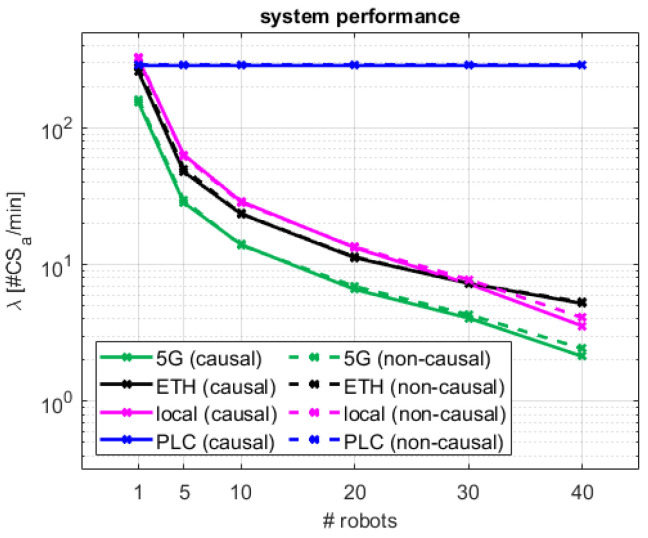
System performance considering median number of cobot actions per minute for the different network configurations.

**Figure 11 sensors-24-05382-f011:**
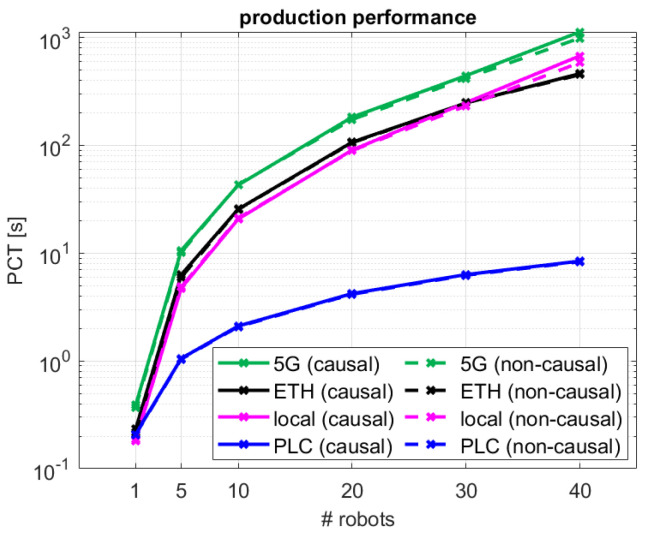
System performance in terms of median production cycle time considering that each cobot performs a single action for the different network configurations.

**Figure 12 sensors-24-05382-f012:**
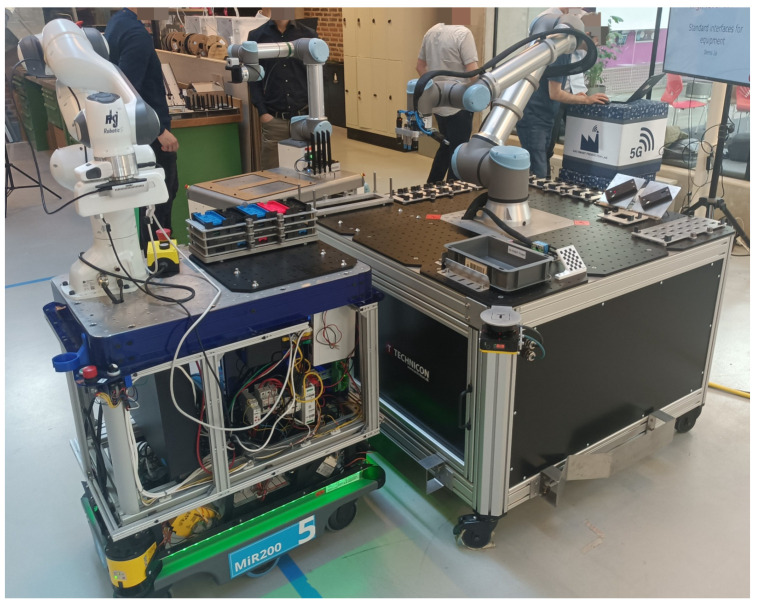
Picture of one of the operational validation scenarios considering coordinated engagement of a stationary robotic cell and two mobile cobots based on the proposed decentralized synchronization method over 5G technology.

**Table 1 sensors-24-05382-t001:** Overview of the technical specifications of the testbed nodes.

Node	CPU	# Cores	RAM
Local machine	Intel^®^ Core™ i9-9900 CPU @ 3.10 GHz	16	32 GB
Factory machine	Broadcom BCM2711, Cortex-A72 (ARM v8) 64-bit SoC @ 1.8 GHz	4	2 GB
MS Azure Cloud	Intel(R) Xeon(R) Platinum 8370C CPU @ 2.80 GHz	1	1 GB

**Table 2 sensors-24-05382-t002:** Overview of the different access network characteristics.

Network	US Data-Rate	DS Data-Rate	RTT Global Cloud	RTT Factory Machine
5G	62.4 Mbps	174.9 Mbps	52.3 ms	-
Ethernet	663.8 Mbps	724.7 Mbps	14.5 ms	0.7 ms

**Table 3 sensors-24-05382-t003:** Structured overview of the different network and computing components considered in each of the examined configurations.

Configuration	Cobot	Network	Access	Line	Multicast
	**Controller**	**Technology**	**Scheme**	**Controller**	**Broker**
5G		Wireless 5G + Internet	Decentralized	N/A	Global cloud
ETH	local	Cabled Ethernet-LAN + Internet		N/A	Global cloud
local	machine	Cabled Ethernet-LAN		N/A	Factory machine
PLC		Cabled Ethernet-LAN	Centralized	Factory machine	N/A

**Table 4 sensors-24-05382-t004:** Summary of the total access time performance of the decentralized causal coordination scheme over the 5G, ETH, and local configurations.

$t_{t}$ [s]	5G			ETH			Local		
# Robots	min	med	99%-ile	min	med	99%-ile	min	med	99%-ile
1	0.2	0.4	0.6	0.2	0.2	0.4	0.2	0.2	0.2
5	0.3	1.8	2.3	0.2	1.1	1.6	0.2	0.8	1.2
10	0.4	4.1	5.0	0.3	2.4	3.1	0.2	1.8	2.2
20	0.5	9.3	10.2	0.3	5.3	6.2	0.2	4.6	5.0
30	0.5	15.6	18.1	0.4	8.4	9.9	0.2	8.5	9.8
40	1.8	30.3	38.3	0.3	11.9	13.3	0.8	17.8	18.5

**Table 5 sensors-24-05382-t005:** Summary of the access efficiency and estimated industrial production cycle times achieved by the decentralized causal coordination scheme over the different configurations for different robotic cell sizes.

	λ [$\#{\mathrm{CS}}_{a}$/min]				*PCT* [s]			
# Robots	5G	ETH	Local	PLC	5G	ETH	Local	PLC
1	154	257	318	285	0.4	0.2	0.2	0.2
5	28	48	63	285	10.6	6.3	4.6	0.4
10	14	23	29	285	42.7	25.6	20.7	2.1
20	7	11	13	285	182.4	106.9	90.3	4.3
30	4	7	7	285	442.6	247.3	247.3	6.3
40	2	5	4	285	1126.8	462.0	675.1	8.4

## Data Availability

Datasets are available on request from the authors.

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
