# Peer review of "Decentralized System Synchronization among Collaborative Robots via 5G Technology"

_sensors, 2024, doi:10.3390/s24165382_

Round 1

Reviewer 1 Report

Comments and Suggestions for Authors

See the attached comments, please

Author Response

General Comments: In this manuscript, the author ‘s wrote on a research topic titled: ‘Decentralized System Synchronization among Collaborative Robots via 5G Technology’. This topic is current and the contributions to knowledge are clearly outlined and demonstrated by the authors through their presented robust methodological algorithms and evidential results, thus make their works novel. The introduction is robust with enough background information to under the overall research approach. The references are relevant and current. The methods adopted are also adequately presented.

Response to General Comments: We thank the reviewer for the time spent in reviewing our work and for the positive and constructive feedback provided. Please, find detailed response to the specific comments below. Revisions/corrections are highlighted in red in the revised version of the manuscript.

Comment 1: While previous works built on cable networks, your decentralised system is 5G technology. Have you also consider the limitations of 5G system network technology and how it can affect or limit the performance of your proposed decentralised system sychronisation among collaborative Robotics. One of it  is the concern is encryption. While apps on 5G networks are encrypted, the 5G NR standard doesn't have end-to-end encryption, leaving it open to certain kinds of attacks. 

Response 1: We thank the reviewer for pointing out these 3 relevant aspects. We have written a discussion about the potential limitations addressing 5G robustness, cloud availability in Section 1, and briefly addressed security/data encryption in the new Section 5. It should be noted that cybersecurity is a topic out of the field of expertise of the authors.

Comment 2: One good thing about cable networks is their high speed compared to wireless cellular systems wherein the 5G network is structured.  I suggest the authors provide to compared the speed of the proposed system compared to the previous ones in references [5-8]. This is in connection with your observed result presentation in line 485-486: ‘observed performance over public 5G is approximately 40% slower as compared 485 to the same solution over cabled Ethernet Internet access’

Response 2: As suggested, we have enhanced the discussions around the performance results in Section 4.1 with selected comparisons to the reported state of the art. 

Comment 3: What is the connection between equations (5) and (6) and how does it applied to your proposed Decentralized System 

Response 3: lambda, in (5), (6) in the updated manuscript, provides a reference of the instantaneous capacity of the robotic cell; while PCT, in (6), (7) in the updated manuscript, provides an estimation of the total cycle execution time assuming a robotic cell where all robots are configured to execute one access. Therefore, both parameters are inversely proportional: for large capacity systems, the resulting cycle time will be small. This has a further implication on industrial production throughput, as fast cycle times are directly related to high production levels. This has been clarified accordingly in the text at the end of Section 3.

Reviewer 2 Report

Comments and Suggestions for Authors

In this paper, the synchronization and coordination of a decentralized system of 5G technology is studied to realize the synchronization and coordination of a decentralized system containing 10 robots. This article needs to be properly modified before it may be accepted for publication. Please note that the following questions are raised here.

1.     For each of your formulas, I think it is necessary to explain the symbols involved in the formula.

2.     The English and logicality of the whole thesis need to be improved.

3.     In the second section of this paper, you propose a decentralized synchronization coordination scheme, but there is a lack of relevant theoretical formula analysis on the system.

4.     In your performance diagram, you need to specify the meaning of different lines in the box diagram.

5.     On the comparison of non-causal and causal relationship, it may be more obvious to put them together for comparison.

6.     In Figure 8, there is a lack of comparison with traditional centralized methods.

7.     Why when the number of robots in the system exceeds 10, the effect of the distributed system is not good, and there is a lack of analysis.

8.     The title of 4.2 is ' Causal vs. Non-causal Decentralized Access', why don 't you compare the two orders in Figure 9?

9.     This article lacks a detailed analysis of Figure 10.

10.  This article lacks a coherent exposition of ' further research prospects, advantages and disadvantages of this study '.

Author Response

General Comments: In this paper, the synchronization and coordination of a decentralized system of 5G technology is studied to realize the synchronization and coordination of a decentralized system containing 10 robots. This article needs to be properly modified before it may be accepted for publication. Please note that the following questions are raised here.

Response to General Comments: We thank the reviewer for the time spent in reviewing our work and for the constructive feedback provided. Please, find detailed response to the specific comments below. Revisions/corrections are highlighted in red in the revised version of the manuscript.

Comment 1: For each of your formulas, I think it is necessary to explain the symbols involved in the formula.

Response 1: We have carefully revised the algorithm in Section 2 and formulas in Section 3 and added and explained all missing symbols in the text. 

Comment 2: The English and logicality of the whole thesis need to be improved.

Response 2: The manuscript has been revised by one native speaker prior to submission. Further, we would like point that one of the reviewers indicated “English language fine. No issues detected”, while the two others expressed “I am not qualified to assess the quality of English in this paper”. The English language has been thoroughly checked in the updated version of the manuscript.

Comment 3: In the second section of this paper, you propose a decentralized synchronization coordination scheme, but there is a lack of relevant theoretical formula analysis on the system.

Response 3: We thank the reviewer for pointing this aspect out. To address it, we have computed the time and space complexity of the proposed algorithm. Overall, the algorithm has a time complexity of O(n^2) and a space complexity of O(n). We have added explanatory text in Section 2.

Comment 4: In your performance diagram, you need to specify the meaning of different lines in the box diagram.

Response 4: No specific aspect has been taken. Boxplots settings and meanings are fully specified within the current version of the manuscript in Section 4.1 (page 10): “To ease the understanding, each set of results is color-coded following the same color schemes used in Fig. 3 for each of the access topologies: 5G in green, ETH in black, local in magenta. The different sub-figures summarize the statistics of t_a and t_r in the shape of boxplots where the boxes indicate performance results bounded within the 25-75%-iles, and the middle line indicates the median value of the distributions (50%-ile). The lower and upper whiskers indicate values at 1%-ile and 99%-ile, respectively.”

Comment 5: On the comparison of non-causal and causal relationship, it may be more obvious to put them together for comparison.
Response 5: We thank the reviewer for this observation. Prior to submission, we considered several different ways of representing the performance data. As, in general, the performance in the causal and non-causal cases is very similar, with a small average difference of 0.1-4.2 seconds, there was no obvious way of overlapping their graphical representation without incurring in complex figure with an extensive colour palette and excessive number of lines/elements. We have updated Figure 8 and provided further input on the Δ offset computation plotted in Figure 9 (which quantifies the average performance difference between causal and non-causal access and is now defined in Equation 5).  We strongly believe that the current representations in Figures 5, 6, 7, 8 and 9, together with the increased details through Sections 4.1 and 4.2 should suffice for the required comparison. 

Comment 6: In Figure 8, there is a lack of comparison with traditional centralized methods.

Response 6: The traditional centralized methods are included in the figure by means of the PLC survival time reference. The corresponding discussion within Section 4.1 has been enhanced. Further, we have updated the figure and its inline description to include both coordinated causal access and uncoordinated non-causal access performance for the sake of completeness. 

Comment 7: Why when the number of robots in the system exceeds 10, the effect of the distributed system is not good, and there is a lack of analysis.

Response 7: We thank the reviewer for pointing this aspect out. We have included this discussion in perspective of Figure 8 in Section 4.1.

Comment 8: The title of 4.2 is ' Causal vs. Non-causal Decentralized Access', why don 't you compare the two orders in Figure 9?

Response 8: We thank the reviewer for noticing this. It is not possible to compare the penalty in execution time for both orders, as, actually, the overhead cost is estimated from their average difference in absolute execution time values. We have added a clarification in Section 4.2, which we believe is further enhanced by the definition of the parameter Δ and related texts added in Section 3.1.

Comment 9: This article lacks a detailed analysis of Figure 10.

Response 9: We acknowledge the comment of the reviewer. We have extended the analysis of Figure 10 in Section 4.3.

Comment 10:  This article lacks a coherent exposition of ' further research prospects, advantages and disadvantages of this study '.

Response 10: We acknowledge the comment of the reviewer. We have added a new Section 5, where we briefly address these aspects together with those about specific validation in operational conditions.

Reviewer 3 Report

Comments and Suggestions for Authors

Dear Authors

The subject of the publication fits into current trends related to decentralized control, especially based on wireless technologies (in this case, applications with collaborative robots) and from this aspect I evaluate it positively.

Division of the content of the publication into individual chapters (summary, analysis of the current state of technology, identification of the research area and proposed solution to the assumed goal (including: decentralized synchronization algorithm, construction of a test application, experimental research, along with the development of efficiency assessment indicators) and preparation of the results of performance experiments for I rate the developed network architecture diagrams (wired/wireless) highly, and the conclusions are correct and can constitute a starting point for further research in this field.

From my point of view, there is some dissatisfaction with the authors' not very clear (precise) definition: for what number of robots (cobots) were tests carried out in real conditions (experimental station)? - the text (including Figures 1-2) shows that the station was equipped with two cobots, and the trials and tests that follow concern up to 30 units? - I think this should be clearly presented in the publication.

Generally, I think that the publication is of a good standard, I like it and I recommend it for publication.

Author Response

General Comments:  The subject of the publication fits into current trends related to decentralized control, especially based on wireless technologies (in this case, applications with collaborative robots) and from this aspect I evaluate it positively. Division of the content of the publication into individual chapters (summary, analysis of the current state of technology, identification of the research area and proposed solution to the assumed goal (including: decentralized synchronization algorithm, construction of a test application, experimental research, along with the development of efficiency assessment indicators) and preparation of the results of performance experiments for I rate the developed network architecture diagrams (wired/wireless) highly, and the conclusions are correct and can constitute a starting point for further research in this field. Generally, I think that the publication is of a good standard, I like it and I recommend it for publication.

Response to General Comments: We thank the reviewer for the time spent in reviewing our work and for the positive provided. Please, find detailed response to the specific comments below. Revisions/corrections are highlighted in red in the revised version of the manuscript.

Comment 1: From my point of view, there is some dissatisfaction with the authors' not very clear (precise) definition: for what number of robots (cobots) were tests carried out in real conditions (experimental station)? - the text (including Figures 1-2) shows that the station was equipped with two cobots, and the trials and tests that follow concern up to 30 units? - I think this should be clearly presented in the publication.

Response 1: We thank the reviewer for pointing this aspect out. We have added some details about this in the new Section 5.

Round 2

Reviewer 2 Report

Comments and Suggestions for Authors

The article can be accepted by the journal now.